# Impact of a 24-Week Workplace Physical Activity Program on Oxidative Stress Markers, Metabolic Health, and Physical Fitness: A Pilot Study in a Real-World Academic Setting

**DOI:** 10.3390/jfmk10030348

**Published:** 2025-09-12

**Authors:** Gabriele Maisto, Maria Scatigna, Simona Delle Monache, Maria Francesca Coppolino, Lorenzo Pugliese, Anna Maria Sponta, Loreta Tobia, Elio Tolli, Pierfrancesco Zito, Valerio Bonavolontà, Leila Fabiani, Chiara Tuccella, Maria Giulia Vinciguerra

**Affiliations:** 1Department of Biotechnological and Applied Clinical Sciences, University of L’Aquila, 67100 L’Aquila, Italy; gabriele.maisto@graduate.univaq.it (G.M.); simona.dellemonache@univaq.it (S.D.M.); lorenzo.pugliese@univaq.it (L.P.); pierfrancesco.zito@univaq.it (P.Z.); chiara.tuccella1@student.univaq.it (C.T.); mariagiulia.vinciguerra@univaq.it (M.G.V.); 2Department of Life, Health and Environmental Sciences, University of L’Aquila, 67100 L’Aquila, Italy; maria.scatigna@univaq.it (M.S.); mariafrancesca.coppolino@univaq.it (M.F.C.); annamaria.sponta@univaq.it (A.M.S.); loreta.tobia@univaq.it (L.T.); elio.tolli@graduate.univaq.it (E.T.); leila.fabiani@univaq.it (L.F.); 3Department of Neurosciences, Biomedicine and Movement Sciences, University of Verona, 37131 Verona, Italy

**Keywords:** oxidative stress, workplace physical activity, wellbeing, physical fitness, employees, redox status, exercise, biochemical parameters, statin

## Abstract

**Background**: Previous studies showed that workplace physical activity programs (WPAPs) could improve general health among employees. However, there is a lack of correlation between oxidative redox status and the metabolic and physical fitness (PF) of workers. The objective of the study was to evaluate the improvements of a 24-week combined circuit training and mobility training program on PF, oxidative redox status, and metabolic parameters on healthy academic employees. **Methods**: Twenty-six university employees (52.8 ± 11.5 years) followed a 24-week WPAP composed of two circuit training sessions and one mobility training session per week. PF components were assessed through one leg stand, shoulder/neck mobility, handgrip, dynamic sit-up, jump and reach, and 2-Minute step test (2MST). Oxidative stress and antioxidant potential were evaluated through derived-Reactive Oxygen Metabolites (d-ROM) and biological antioxidant potential (BAP) tests, respectively. Metabolic measurements included total cholesterol, LDL-C, HDL-C, triglycerides, and fasting plasma glucose. All assessments were conducted at baseline and after 24 weeks. **Results**: D-ROM values increased significantly likely due to an acute adaptive response to exercise and a stable BAP/d-ROM ratio was maintained. At baseline, subjects with higher 2MST scores showed a better BAP/d-ROM ratio compared to those with lower 2MST scores, which was also associated with normal weight status (*p* < 0.05), healthy values of triglycerides (*p* < 0.01), and LDL-C (*p* < 0.01). Excluding statin-treated subjects, an intriguing shift toward a condition of enhanced antioxidant capacity was observed. **Conclusions**: Overall, the 24-week WPAP improved metabolic health and maintained redox balance, despite increased reactive oxygen species (ROS) production. Statin supplementation may have hidden antioxidant adaptations to physical exercise, an intriguing observation that warrants further studies.

## 1. Introduction

The practice of physical activity (PA) and the adoption of a healthy lifestyle can improve people’s quality of life (QoL), preventing most chronic non-communicable diseases (NCDs) and promoting a successful aging process [1,2].

Physical inactivity and sedentary behaviors are major factors of mortality and morbidity at the global level [3]. Studies indicate that employees spend 75% of their worktime in sedentary activities [4,5,6,7]. Higher sedentary time is associated with a greater incidence risk for cardiovascular diseases, diabetes, and cancer [8], specifically in the working population [9,10]. Indeed, many companies have adopted strategies to encourage their employees to exercise and adopt healthy lifestyles, as the workplace seems to be the ideal environment to be physically active and reduce sedentary behaviors [11,12].

The implementation of physical exercise programs within companies has been associated with many benefits, including improvements of QoL, mood, relationships between employer and employees, the workplace environment, productivity and reducing the risk of developing NCDs, musculoskeletal disorders, burnout, all-cause mortality, absenteeism, and frequency of errors [13,14,15,16,17].

By experiencing a workplace physical activity program (WPAP), employees improve their awareness about the advantages of healthy choices and are encouraged to maintain active behaviors in their lives [18]. Such increased physical activity may have positive effects on biomarkers of risk for cardiovascular disease and oxidative stress [19,20]. A few studies showed that WPAP affects these variables in generally healthy employees, in particular blood pressure and blood lipid levels [21]. Previous longitudinal studies have demonstrated the effectiveness of WPAP interventions in improving blood lipid profile [22].

As with other stimuli in the body, physical activity can physiologically produce a moderate increase in free radicals that can alter the balance between oxidant and antioxidant species by increasing oxidative stress and lipid peroxidation [23,24]. In turn, free radicals can activate molecular mechanisms in the cell to protect and adapt it to oxidative stress and can enhance immunological defense [25].

An increase in antioxidant potential following appropriate physical activity has been demonstrated [26,27], as well as a suppression of oxidative stress following a gradual increase in the intensity of physical activity [28]. The status of increased oxidative stress combined with reduced antioxidant potential has a strong association with the development of certain diseases [29,30]. An intracellular proinflammatory cascade is potentially initiated via ROS, triggering the development of inflammation, which in turn can exacerbate oxidative stress, creating a vicious cycle [31]. However, the balance between increase in oxidative stress and antioxidant potential resulting from adequate physical activity boosts immunity [32,33], confirming the role of regular and prolonged physical activity in preventing the onset of heart disease, stroke, diabetes, lung and bone diseases, and cancer [34,35]. For this reason, the process of validating biomarkers of oxidative stress in clinical settings has received a great deal of attention in recent years [36,37]. According to a recent review, regular training seems to have adaptations that can reduce exercise-related oxidative stress and it has been suggested that exercise-induced oxidative stress depends on its intensity; however, heterogeneity between studies, different exercise protocols, and monitoring of individual antioxidant enzyme biomarkers make it difficult to obtain conclusive data [38].

There are few physical activity interventions in the workplace setting, and even fewer studies have analyzed the oxidative stress status of healthy employees [39]. Despite the well-known benefits of WPAPs, their impact on oxidative stress status using the BAP/d-ROM ratio, in a real-world academic setting, remains scarcely investigated. Furthermore, the use of pharmacological treatments such as statins can influence oxidative stress status. A significant increase in the circulating antioxidant enzyme concentrations has been associated with statin treatment [40]. However, it is unclear if the concomitant use of statins may interact with the physiological adaptation in oxidative stress status induced by exercise, that is, an amplifying or attenuating antioxidant effect. It is then of paramount interest to verify if such programs can effectively impact oxidative stress status and the physical fitness of employees. In 2016, the University of L’Aquila started an on-site physical exercise program intended for all employees and post-lauream students, called ‘University on the Move’, aimed at promoting health by increasing the level of physical activity. Therefore, the aim of this study was to verify the impact of a 24-week WPAP intervention on metabolic biomarkers, oxidative redox status, physical fitness, and lifestyle components in an academic community.

Whereas many previous studies adopted an informational and attitudinal type of intervention, in this research, a concrete and structured on-site physical activity program, harmonized in the physical and organizational university context, was offered to the employees.

## 2. Materials and Methods

### 2.1. Study Design

The study is based on a quasi-experimental design with a longitudinal single-sample evaluation (one-group pre-test–post-test) at baseline (T0) and after 24 weeks (T1) (Figure S3).

### 2.2. Sample

The non-probabilistic sample consisted of twenty-six employees (5 men and 21 women) from the University of L’Aquila (L'Aquila, Italy) who voluntarily enrolled. The study was conducted in accordance with the Declaration of Helsinki and approved by the Internal Review Board of the University of L’Aquila (opinion of 28 May 2020, no. 17/2020). All participants provided written and signed informed consent.

### 2.3. Inclusion and Exclusion Criteria

Inclusion criteria were (1) full-time employment at the University of L’Aquila, (2) age between 23 and 65, and (3) medical clearance for PA. Exclusion criteria included (1) diagnosis of cardiovascular, respiratory, or metabolic diseases and (2) musculoskeletal conditions limiting exercise participation.

### 2.4. Intervention

The WPAP consisted of two circuit training sessions and one mobility exercise session per week, on non-consecutive days, for 24 weeks. All sessions were supervised by the research staff and started with a comprehensive 10 min warm-up that incorporated dynamic stretching sequences and progressive aerobic exercises to prepare participants for the main workout phase. Every session ended with a 10 min cool-down protocol, incorporating static stretching, relaxation, and breathing techniques.

The central part of the circuit training session (Appendix A) lasted 24 min and comprised eight stations, repeated for three rounds, following a work/rest ratio of 1:1 (30 s of exercise, 30 s of rest). The implemented circuits and their exercises were not fixed throughout the WPAP but were chosen by the research team every month. In Appendix A, numbers indicate the exercise stations that targeted the four physical fitness components: cardiorespiratory fitness, muscular strength, muscular endurance, flexibility, and balance. Apart from the flexibility and balance component, which always had only one exercise, every two weeks, a specific component was emphasized by adding one extra station (for a total of three exercises).

Appendix A provides the first three circuits of the WPAP. The first circuit (Circuit1), also shown in Appendix A, was implemented in weeks 1 and 2 of the intervention and included three exercises to improve cardiorespiratory fitness (CRF), as indicated in Appendix A. In Circuit2, the focus moved on to the muscular strength component and in Circuit3, the focus was on the muscular endurance component. The attendance of the WPAP was 55%, with no adverse events registered.

The mobility session included yoga and Pilates exercises, with sequences from various positions (e.g., upright, supine, prone, and lying on the side) using body weight and elastic bands. Each exercise lasted 15–30 s or for 12–20 repetitions with 2–4 sets as needed (Appendix A).

To maintain training effectiveness, training load progressively increased throughout the WPAP period, according to ACSM guidelines [41]. Specifically, several progression rules were used. The circuit program focused on a specific PF component every two weeks, following a fixed rotation scheme (Appendix A). Progression is based on training the components of physical fitness through circuit training, as stated in several studies. Furthermore, exercise complexity was gradually increased across the WPAP. For instance, while Circuit1 (Appendix A) used the squat, Circuit 3 progressed to the jump squat, adding complexity and therefore increasing the intensity of the stimulus (Appendix A). In addition, the minimum number of repetitions to accomplish for each station was set higher every four weeks and progression in resistance was ensured by providing participants progressively heavier dumbbells, medicine balls, kettlebells, and resistance bands. The general list of the equipment used for all the training sessions is provided in Appendix A.

The Borg Rating of Perceived Exertion Scale of 6–20 [42] was used to monitor the exercise intensity, which was maintained in the range of 13–15, corresponding to moderate-to-vigorous intensity [43]. For every training session, rate of perceived exertion (RPE) values were recorded once, at the end of the central part. RPE compliance was 96%, while RPE values within the target range of 13–15 were 71%.

### 2.5. Outcome Measures

The evaluation protocol included laboratory analyses of blood samples to assess the oxidative status variables as the primary outcome (d-ROMs, BAP, and BAP/d-ROMs) and other biochemical parameters as secondary outcomes related to metabolic risk profiling (blood lipids, fasting glucose). Moreover, instrumental measurements, field tests, and questionnaires allowed us to describe the sample and to add information on the further impact of the intervention. To explore the potential role of statins in modulating oxidative stress, a longitudinal analysis on a pooled sample compared mean values of pro-oxidant status (d-ROMs) and antioxidant status (BAP) and the derived oxidative stress index (BAP/d-ROMs) at baseline (T0) and after the intervention (T1).

### 2.6. Oxidative Stress

The oxidative and antioxidant status markers were assessed using the Free Carpe Diem device (FREE^®^ Carpe Diem; Diacron International, Grosseto, Italy https://diacron.com/). Blood samples were always taken in the morning (before 12:30 AM), in a fasting state, and at least 48 h after the last training session. The system measures hydroperoxides in plasma using the Diacron reactive oxygen metabolite (d-ROM) test. When iron is present, d-ROM levels in plasma can generate alkoxyl and peroxyl radicals. The inter-assay coefficients of variability (CV) were 1.79% for the d-ROM test and 3.05% for the BAP test, according to the manufacturer’s data sheets.

These radicals can oxidize an alkyl-substituted aromatic amine, transforming it into a pink-hued derivative. This transformation can be quantified spectrophotometrically at a wavelength of 546 nm. The outcomes are reported in arbitrary units known as Carratelli units (U.CARR), where one U.CARR is equivalent to 0.08 mg of H_2_O_2_ per 100 mL, as proposed by Iorio & Balestrieri, 2009 [44]. The test on the biological antioxidant activity of plasma (BAP) utilizes a colored solution of ferric ions (Fe^3+^) bound to a chromogenic substrate, which undergoes decolorization when Fe^3+^ is reduced to Fe2+ by the antioxidant capacity present in the plasma. The degree of decolorization is measured photometrically at a wavelength of 505 nm. The normal BAP value in healthy individuals is greater than 2200 μmol/L, as proposed by Iorio & Balestrieri, 2009 [44,45,46] (Appendix A). The variables used for the outcome analysis are d-ROMs, BAP, and the ratio of BAP to d-ROMs (index of oxidative stress), as proposed in previous studies [21,47,48,49].

### 2.7. Anthropometry, Metabolic, and Behavioral Variables

Anthropometric measurements included height, weight, and Body Mass Index (BMI) (kg/m^2^). An anamnestic questionnaire collected information about lifestyle habits, drug and dietary supplement use, and physical activity practice. Biochemical parameters included total cholesterol, high-density lipoprotein cholesterol (HDL-C), low-density lipoprotein cholesterol (LDL-C), triglycerides, and fasting plasma glucose. Samples were collected after at least 12 h of fasting. Blood samples were collected in 10 mL EDTA, sodium heparin, and serum separator vacuum tubes (Vacutainer, DB Italia, Milan, Italy). All serum samples were allowed to clot, then serum and plasma were separated by centrifugation at 4 °C for 15 min at 2000× *g*. After the removal of the plasma and the buffy coat, the erythrocytes were washed and then lysed with cold distilled water.

### 2.8. Physical Fitness

The following tests from the validated AlphaFit Test Battery [50] were used to assess physical fitness components: the one leg stand test was utilized to assess balance, the shoulder/neck mobility test was used to verify scapulohumeral mobility, the handgrip test measured participants’ upper limb and grip strength, the sit-up test was used to assess core endurance, the jump and reach test was used to evaluate lower limb power. In addition, the 2 min Step Test was administered to assess cardiorespiratory fitness [51].

### 2.9. Statistical Analysis

Statistical analysis was performed using STATA/BE 17.0 software by StataCorp LLC, College Station, TX, USA.

Frequencies were calculated for categorical variables, and tests of association were used for unpaired (Chi-squared test, with Fisher’s correction) and paired groups (McNemar’s test, symmetry test). For quantitative variables, measures of central tendency and variability were used (arithmetic mean, standard deviation, min–max range of variability) and, after checking for the absence of normality in the distribution of the variables (Shapiro–Wilk test), the statistical significance of differences was determined using non-parametric tests for unpaired data (Mann–Whitney rank-sum test, Cuzick’s trend test) and for paired data (Wilcoxon matched-pairs signed-rank test). The absolute change Δ with 95% IC and the effect size, Wilcoxon’s r, for primary outcome variables were calculated. Spearman’s Rho was tested as an ordinal correlation estimate. Tests were two-tailed, and *p*-values of less than 0.05 were considered statistically significant.

## 3. Results

### 3.1. Sociodemographic Characteristics and Health Habits

At baseline, the participants had an average age of 52.8 ± 11.5 years (range: 21–65 years). As regards the professional position, 53.9% of them were academic staff, with the remainder comprising technical-administrative personnel (38.4%) and pre- and post-graduate students (7.7%).

In total, 92.3% of the sample does some physical activity on a regular basis (sport and/or physical exercise and/or recreational), with modestly higher levels in men (100.0% vs. 90.5% of women, n.s.), also in terms of weekly frequency (respectively, 3.2 vs. 2.9 times a week, n.s.) and total minutes per week (respectively, 228.0 vs. 195.8 total minutes per week, n.s.) (Table 1).

The most frequently self-reported clinical conditions are dyslipidemia (28.6% of women and 80.0% of men, *p* < 0.05), hypertension (28.6% of women and 20.0% of men, n.s.), and thyroid disorders (28.6% of women and 0.0% of men; less than 10% of women and no men suffered from cancer, diabetes, or heart diseases). Among women, 46.2% are in menopause. Among the women, 42.9% are on medication, compared to 100% of the men (*p* < 0.05) with an average of 0.8 and 1 medication per person, respectively (n.s.) (Table 1). The most used medications are antihypertensives (in five cases), hypolipidemic and, less frequently, anti-inflammatory, antidiabetics, hormones, cardioaspirin, and other drugs used by the individual (e.g., for the treatment of autoimmune diseases, antiacids, etc.). Moreover, 66.7% of women and 40% of men use dietary supplements (n.s.), including source preparations of minerals—e.g., Mg and K—(in eight cases), vitamin D (in seven cases), B-complex vitamins (in six cases), antioxidants—vitamin C, vitamins A and e—(in five cases), iron (in three cases), or other—e.g., probiotics, immunostimulants—(in two cases) (Table 1).

### 3.2. Metabolic and Physical Fitness

The average BMI fell within the normal weight range according to international cut-off values [52] without any significant change between T0 and T1, both among women (respectively, 24.9 ± 4.4 kg/m^2^ and 24.8 ± 4.2 kg/m^2^) and among men (respectively, 25.3 ± 3.6 kg/m^2^ and 25.1 ± 3.3 kg/m^2^) (Table 2). At T0, 57.1% of women and 20.0% of men had total cholesterol plasma concentrations above the normal range for the adult population, and these proportions decreased, respectively, to 28.6% (*p* < 0.05) and 0.0% at T1 (n.s.).

Similarly, the LDL cholesterol levels decreased from 57.1% at T0 to 23.8% at T1 among women (*p* < 0.01) and from 20.0% to 0.0% among men (n.s.). In both times of observation, only 4.8% of women had HDL cholesterol levels outside the normal range (either elevated or reduced), and none of the men did. Plasmatic triglyceride concentration was above the normal limit in 9.5% of women and in 20.0% of men at baseline, and these proportions changed over time in both groups, but not significantly, respectively, to 4.8% (n.s.) and 40.0% (n.s.). At baseline, fasting glucose levels were above the normal range in 9.5% of women and decreased to 0.0% at T1 (n.s.) while none of the men had abnormal values at any time (Table 2).

As shown in Table 3, at baseline, 53.3% of women and 60.0% of men scored ‘high’ performance on the balance test (one leg stand), and over time, these proportions increased in the men to 100.0% at T1, respectively, although without statistical significance. For the shoulder/neck mobility test, fewer than half of the subjects showed ‘no’ or ‘slight restriction’ in movements in both groups (40.0% women and men) across time (from T0 to T1). The handgrip strength test scores fall within the reference range for age and sex at both times, showing a slight, but not significant improvement at T1 (in average, from 28.7 ± 6.1 kg at T0 to 30.0 ± 6.6 kg at T1 in women and from 48.0 ± 5.9 kg to 50.6 ± 11.0 kg in men). At baseline, 60.0% of women and 40.0% of men reported a high score on the jump and reach test, with a slight decrease (to 53.3%) in the women and an improvement in the men (to 80.0%) at T1 but not at a statistically significant level.

Core endurance, measured via sit-up test performance, tends to improve among women, with the proportion reaching high fitness levels increasing from 66.7% at baseline to 80.0% at T1 (difference not statistically significant); in the men, the proportion remained constant at 100.0% across all time points.

The step test showed mild improvements over time, particularly among men: all of them consistently performed above the risk threshold at every time point, and the number of steps increased from 87.6 ± 16.2 at T0 to 100.0 ± 21.8 at T1 (n.s. *p* = 0.0796); among women, the proportion classified as the ‘at risk’ category increased from 6.7% at baseline to 13.3% at T1, but the number of steps also increased from 82.7 ± 16.8 at T0 to 85.5 ± 19.0 at T1 (the differences in both cases were not statistically significant) (Table 3).

Given that over 90% of participants reported regular physical activity at baseline, ceiling effects should be considered when interpreting changes in fitness levels and lipid profiles, as limited room for improvement may attenuate observable effects.

### 3.3. Oxidative Stress

#### 3.3.1. Status of Employees over Time

Pro-oxidant levels, as measured by the d-ROM test, increased significantly in women from 383.2 ± 79.3 to 426.8 ± 87.3 U.CARR (medium effect size: Wilcoxon’s r = −0.435; Δ = 43.6 U.CARR, 95% CIs [18.6, 68.8 U.CARR], *p* = 0.0033), while the increase in men was not statistically significant from 338.4 ± 38.9 to 388.0 ± 66.4 U.CARR (large effect size Wilcoxon’s r = −0.554; average Δ = 49.6 U.CARR, 95% CIs [−2.8, 102.0], *p* = 0.1250). The antioxidant capacity, as measured via the BAP test, showed a mild increase from T0 to T1 in both groups, though it was not statistically significant in women from 2513.8 ± 624.6 to 2640.4 ± 488.2 µmol/L (small effect size: Wilcoxon’s r = −0.067; average Δ = 126.6 µmol/L, 95% CIs [−299.0, 552.2], *p* = 0.6827); in men from 2578.2 ± 943.6 to 2789.2 ± 379.6 µmol/L (small effect size: Wilcoxon’s r = −0.128; average Δ = 211.0, µmol/L 95% CIs [−1095.7, 1517.7], *p* = 0.8125). The BAP/d-ROM ratio decreased slightly in women, from 7.0 ± 2.8 to 6.4 ± 1.6 (small effect size: Wilcoxon’s r = −0.105; average Δ = −0.6, 95% CIs [−1.9, 0.7], *p* = 0.5168); in men, it went from 7.6 ± 2.7 to 7.4 ± 1.5 (small effect size: Wilcoxon’s r = −0.043; average Δ = -0.2, 95% CIs [−3.7, 3.3], *p* = 1.0000), although none of these changes reached statistical significance (Figure 1, Table 4 and Appendix A).

At baseline, a high proportion of participants exhibited elevated levels of oxidative stress: 85.7% among women and 80.0% among men reported d-ROM values ≥300 U.CARR at T0, rising, respectively, to 90.5% and 100.0% at T1. Suboptimal BAP levels (<2200 µmol/L) were observed in 23.8% of women and 20.0% of men at baseline, with a nonsignificant decrease only in men (to 0.0%). According to Iorio’s classification [44], only two different participants (both women) were found to be in an ‘optimal oxidative balance’ condition, one at baseline and the other at T1; 80.0% or more of women and men are classified as ‘potential oxidative stress’ or ‘absolute oxidative stress’ with no statistically significant differences between T0 and T1 observations (Table 5). A shift analysis highlighted that within women, seven (7/21, 33.3%) improved their oxidative stress condition along the study from t0 to t1, moving from a worse category to a better one, five (5/21, 23.8%) worsened it, and nine (9/21, 42.9%) remained in the same category (‘potential oxidative stress’). Instead, within men, only one (1/5, 20.0%) worsened his oxidative stress status, moving from ‘relative’ to ‘potential’ oxidative stress, and four (4/5, 80.0%) remained in the same category (‘potential oxidative stress’).

According to univariate analysis at baseline, men showed more favorable oxidative stress compensation than women (BAP/d-ROM ratio, respectively, 7.57 ± 2.72 vs. 7.00 ± 2.82, Table 4; proportion of ‘absolute oxidative stress’ 0.0% vs. 23.8%, Table 5), but these differences were not found to be statistically significant for Mann–Whitney test and Fisher’s exact test. The average values by age group did not follow a statistical trend .

#### 3.3.2. Association of Oxidative Stress with Metabolic and Other Health Correlates

The univariate analysis on whole sample at T0 (Table 6) revealed a statistically significant association of BAP/d-ROM ratio (as an index of oxidative compensation) with metabolic risk factors such as overweight/obesity, levels of total cholesterol, LDL cholesterol, and triglycerides: the compensation resulted in significantly higher categories below the unhealthy values (i.e., 8.15 ± 2.74 vs. 6.06 ± 2.43 for weight status; 8.71 ± 2.78 vs. 5.51 ± 1.59, *p* < 0.01 for total cholesterol and LDL cholesterol and 8.53 ± 3.10 vs. 7.02 ± 2.63 vs. 4.79 ± 1.12, *p* = 0.0719 for triglycerides). The proportion of subjects with ‘absolute oxidative stress’ is also lower for the different levels of total and LDL cholesterol, but this difference does not reach sufficient statistical significance (7.7% vs. 30.8%, n.s.). For all other variables examined as potential predictors of higher oxidative stress/unbalanced compensation, the indices show less favorable values for the higher-risk categories (i.e., fasting plasma glucose above the normal threshold, smokers vs. ex-smokers/non-smokers, regular alcohol consumption vs. no consumption, supplement use vs. no use) (Table 6).

As shown in Table 6, in the univariate analysis at baseline, a significant inverse correlation was found between the ‘generic’ use of medications and the BAP/d-ROM ratio (6.26 ± 2.65 vs. 8.10 ± 2.64, *p* < 0.05), suggesting that subjects taking drugs, in particular statins, had a lower antioxidative/oxidative stress balance. As shown in Figure 2, we observed a statistically significant increase in ROS levels following the exercise protocol. Specifically, d-ROM levels increased from 381.5 to 438.5 U.CARR (*p* < 0.05), indicating enhanced oxidative stress. Although BAP values also rose from 2522.9 to 2796.3 µmol/L, this change was not statistically significant, suggesting a more limited antioxidant response of this group of subjects. On the other hand, the antioxidative/oxidative stress ratio, which may reflect the balance between antioxidant defense and oxidative stress, showed no significant change (T0: 6.52 ± 1.7 vs. T1: 6.13 ± 3.1 µmol/L/U.CARR; ns), indicating that, despite the increases in ROS, the overall balance of the oxidative state remained constant (Figure 2A). Conversely, as shown in Figure 2B, the analysis of the subgroup not receiving statin therapy did not show a significant change in d-ROM levels between T0 and T1, suggesting a reduced oxidative response to exercise.

**Figure 2 jfmk-10-00348-f002:**
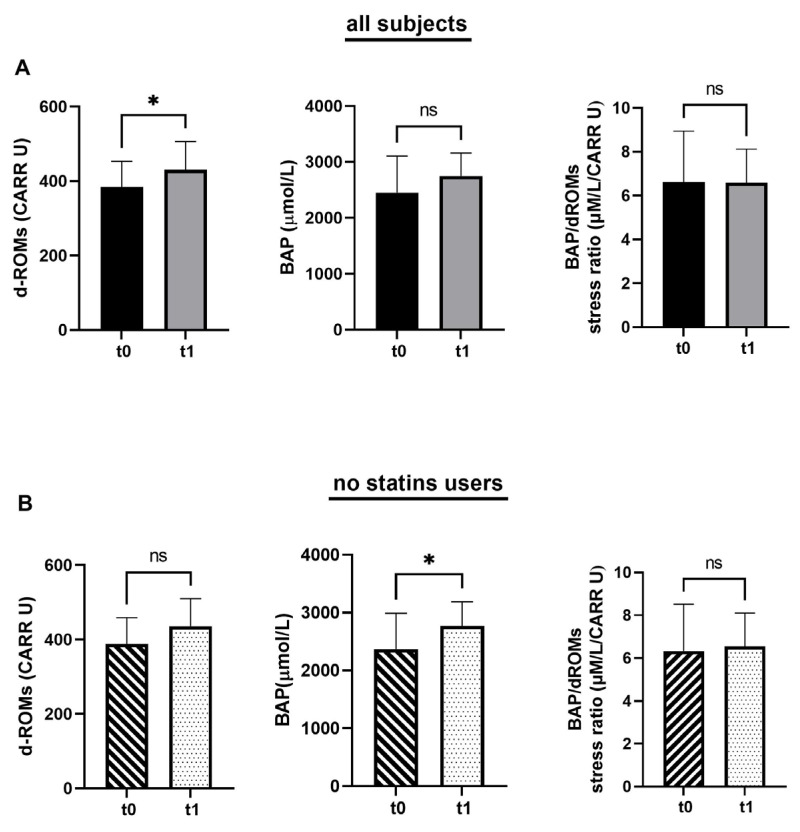
Time-dependent effects of exercise on derivatives of reactive oxidative metabolites (d-ROMs), biological antioxidant potential (BAP), and BAP/d-ROMs, indicating antioxidative/oxidative stress ratio. (**A**) d-ROMs, BAP, and BAP/d-ROMs in whole group (**all subjects**) (*n* = 26). (**B**) d-ROMs, BAP, and BAP/d-ROMs in subgroup excluding statin-treated subjects (**no statin users**) (*n* = 24). Error bars represent standard deviation (±SD). * = Statistically significant; ns = not significant.

On the contrary, BAP levels increased significantly over the same period (*p* < 0.05), highlighting a robust increase in antioxidant capacity, while the BAP/d-ROM ratio did not change significantly.

Concerning physical fitness levels, the univariate analysis overall showed more favorable values for oxidative compensation indices and absolute oxidative risk among more fit employees, even if only in two cases—the dynamic sit-up test and the 2-Minute Step Test—with sufficient or near-sufficient statistical significance. The BAP/d-ROM ratio is higher among subjects with ‘high’ abdominal muscular endurance compared to the ‘low/mid’ category (7.67 ± 2.52 vs. 5.85 ± 3.02, *p* = 0.0668), and the proportion of subjects with ‘absolute oxidative stress’ is lower in the former group than in the latter (11.1% vs. 37.5%, n.s.). The number of repetitions performed in the 2-Minute Step Test is significantly correlated with the BAP/d-ROM ratio; that is, the subject’s compensation level improves as the number of steps performed per unit of time increases (Spearman’s rho = 0.4038, *p* < 0.05). This association is also confirmed by comparing subgroups: the BAP/d-ROM ratio is higher among those who reach or exceed the risk threshold of 65 steps in the test (7.70 ± 2.53 vs. 3.87 ± 1.37, *p* < 0.01), and the proportion of subjects with ‘absolute’ oxidative stress is lower (4.5% vs. 100.0%, *p* < 0.001). Moreover, a significant increasing trend is observed across performance tertiles for the raw ratio (6.13 ± 2.70 vs. 8.06 ± 2.95 vs. 8.11 ± 2.26, *p* = 0.0519) (Table 7).

## 4. Discussion

This study examined the impact of a 24-week WPAP on metabolic markers, oxidative redox status, and physical fitness in an academic community. The intervention, which combined mobility and circuit training sessions, was designed to promote a healthier and more active work environment and correct lifestyles in a context where sedentariness is often prevalent [39].

Overall, our results showed that the 24-week WPAP was associated with a rising trend in antioxidant potential (BAP), with a potentially beneficial increase in the ROS levels in response to the hormetic adaptation to the workout program. In particular, an absolute increase of 43.6 U.CARR in women and of 49.6 U.CARR in men emerged, with an effect size of, respectively, ‘medium’ and ‘large’. Indeed, the BAP/d-ROM ratio remained constant across time, which is in line with previous studies, suggesting that physical exercise contributed to maintaining an oxidative balance and protecting from oxidative stress, as previously reported [50,51,53,54]. In parallel, the intervention seemed to determine a significant reduction in total and LDL cholesterol; in particular, 33.3% of women and 20.0% of men reduced their LDL values below the clinical upper cut-off (130 mg/dL). These findings are consistent with prior results indicating that regular exercise benefits the blood lipid profile [52,55]. Similar reductions have also been reported in a men-only sample [56], or in studies with shorter intervention durations [57]. Other approaches that also integrate dietary modification (lifestyle interventions) seem to achieve larger effects in terms of reduction in LDL cholesterol as well as lower blood pressure values [58].

Glucose, HDL, and triglyceride levels showed a favorable but not statistically significant trend. In addition, improvements were also observed in various physical fitness parameters, especially in CRF.

Numerous factors over the course of a person’s life may influence the overall oxidative balance, either towards a state of unbalanced oxidative stress, caused by a higher production of ROS and a lower production of antioxidants (e.g., age, smoking, stress, drugs, acute exercise, poor nutrition), or towards a state of oxidative balance, where the physiological production of ROS is balanced and counteracted by the intervention of antioxidants. In this sense, exercise is considered one of the beneficial factors of a healthy lifestyle and is now considered an indispensable element of health that is able to reduce the risk of cardiovascular, endocrine, and osteomuscular disorders, such as diseases of the immune system, and the risk of developing cancer [59].

Our results showed improvements in several physical fitness components, especially for handgrip strength and CRF (2 min Step Test). Specifically, CRF in men improved nearly to statistical significance (Table 3).

Thus, although the results did not reach statistical significance, probably due to the small and heterogeneous sample size over time, the trends support the idea that moderate-to-vigorous regular physical activity can maintain and improve several physical fitness components in the workplace [18]. This outcome can foster an active work environment, as suggested by Global Action Plan for Physical Activity (GAPPA) and by the International Society for Physical Activity and Health (ISPAH) statement [60,61].

Moreover, the improvements observed in fitness tests, particularly in the 2-Minute Step Test, support the effectiveness of the program in increasing functional capacity. The positive association between performance in tests (e.g., sit-ups and step tests) and redox compensation parameters suggests that a higher level of fitness correlates with better efficiency of the antioxidant system. The significant association observed exclusively at baseline may be dependent on elevated ROS levels in many participants at that time point, possibly introducing a bias or ceiling effect in the data. Interpretation of changes in physical fitness and lipid profiles should consider potential ceiling effects. At baseline, over 90% of participants reported regular physical activity, indicating a high level of engagement in health-promoting behaviors prior to the WPAP. This high baseline activity level may have limited the capacity for further measurable improvement in both physical fitness and lipid outcomes, thereby attenuating detectable intervention effects. Consequently, observed trends should be interpreted within the context of these baseline characteristics.

At baseline, the subjects who showed higher values, i.e., higher repetitions in the step test, had a higher BAP/d-ROM ratio (Table 7), suggesting a protecting role of the aerobic exercise training on oxidative stress status [62].

Thus, one of the relevant results of this work is the significant increase in ROS levels, particularly relevant in women, which can be interpreted as a physiological response to physical exercise. In fact, exercise, as demonstrated by numerous studies, induces a transient increase in ROS by stimulating the body’s natural antioxidant response [29,33]. In any case, the maintained stability of the BAP/d-ROM ratio suggests the establishment of an ‘eustress’ condition determined by cellular adaptation and not a pathological response [63]. Furthermore, the timing of blood sampling relative to the last training session is a crucial factor [64]. The lack of homogeneity in this factor within the subject group could have contributed to the observed effect on ROS, as physical exercise acutely increases their production in a time-dependent manner [29]. A uniform washout was not enforced, and the small sample size did not allow for a sensitivity analysis. In addition, the observed increase in ROS could be a result of the use of drugs and specific supplements, and not just the training, as demonstrated by the often-conflicting results found in antioxidant intervention studies [65].

In addition, elevated ROS levels at baseline can also indicate a chronic oxidative stress status that is often associated with several factors, such as age, sedentary behavior (habits), and drug use [66,67]. A significant percentage of participants assumed drugs, specifically statins (Table 1); therefore, this variable was considered in the analysis of oxidative stress.

Indeed, as described in the literature, statins can significantly affect plasma cholesterol levels and, indirectly, the body’s response to oxidative stress [68]. For this reason, we first decided to look at oxidative stress levels in the entire sample, because oxidative status, among various factors, does not seem to be particularly dependent on gender [69,70,71,72].

Moreover, to isolate the effect of statins, the subgroup of individuals not taking statins was considered separately, showing a reduced oxidative response to exercise. This lack of response was associated with a significant increase in BAP levels, indicating a substantial increase in antioxidant capacity and suggesting that the redox equilibrium shifted in a beneficial way, leading to a more effective antioxidant response to physical exertion in the absence of statins. Several mechanisms have been proposed to explain this result. Recent studies suggest that statins may influence redox status through the modulation of endogenous antioxidant enzymes such as SOD and GPx, although the underlying mechanism is still incompletely understood [40]. In addition, statins competitively inhibit the enzyme HMG-CoA reductase, reducing the biosynthesis of CoQ10. Reduced CoQ10 availability may limit the regeneration of the antioxidants vitamin C and vitamin E and weaken the ability to counteract the exercise-induced oxidative stress [73]. In this context, statin users could have a lower antioxidant adaptation to exercise. The increase in antioxidant capacity, after excluding the individuals taking statins, supports the hypothesis that CoQ10 depletion is a key mediator of the lowered antioxidant response seen in statin users.

On the other hand, it is also possible that individuals taking statins have underlying cardiovascular conditions that inherently involve higher levels of oxidative stress or inflammation, which could obscure the exercise-induced antioxidant response.

Therefore, these results imply that hypolipidemic drugs may mask the body’s antioxidant response to exercise-induced oxidative stress. However, these subgroup analyses were exploratory and should be interpreted with caution given the small sample size.

The major limitation of the present study is the single-sample, uncontrolled, and non-randomized design. As in other contexts, this quasi-experimental approach has been forced by the first aim of intervention, i.e., a health-promoting program targeted to the entire academic community, and it was ethical to accept the ‘natural’ enrollment of employees interested in improving their health. The absence of a control group that did not receive the intervention makes it difficult to determine whether the observed changes were directly caused by the WPAP or due to unrelated influences. This lack of control also reduces internal validity and weakens the ability to draw causal inferences. To address this limitation, incorporating a randomized control group would allow to more accurately isolate the effects of the intervention. Moreover, the low sample size and poor homogeneity (high number of women out of men, large age range, etc.) worsen statistical power. Some improvement trends are detectable but not at a level of sufficient statistical significance, and it was not possible to apply multivariate statistical analysis models (e.g., multiple regression models) that would have allowed for controlling the confounding variables. Given the multiplicity of outcomes assessed without adjustment, these analyses should be considered exploratory and therefore interpreted with caution. Moreover, because of the uncontrolled before/after design, a potential effect of regression on the mean or seasonality cannot be excluded.

In addition, it is important to emphasize the importance of monitoring participants’ nutrition and diet, and for this reason, we are conducting nutritional surveys for future studies and developments using a food frequency questionnaire administered at each follow-up.

Despite the potential shortcomings, the uncontrolled before/after design provided valuable insights into possible outcomes in real-world settings [74]. Since the primary objective of this research is to verify the real benefits of a WPAP in a workplace context, future studies should have a larger sample, which would allow for better homogeneity of the compared subgroups and increase measured ex post power.

## 5. Conclusions

In conclusion, the WPAP showed a positive impact with notable and encouraging trends, even if some differences were not statistically significant. The intervention appears to have been effective in improving the lipid profile and physical fitness. Despite the increase in ROS, it also appears that a favorable oxidative balance was maintained. Overall, these results underscore the importance of considering the interaction between exercise, pre-existing conditions, and drug therapies in modulating the body’s physiological response. In addition, this study may have uncovered a potential influence of statins on the body’s antioxidant response to exercise, an interesting aspect to investigate through further research.

## Figures and Tables

**Figure 1 jfmk-10-00348-f001:**
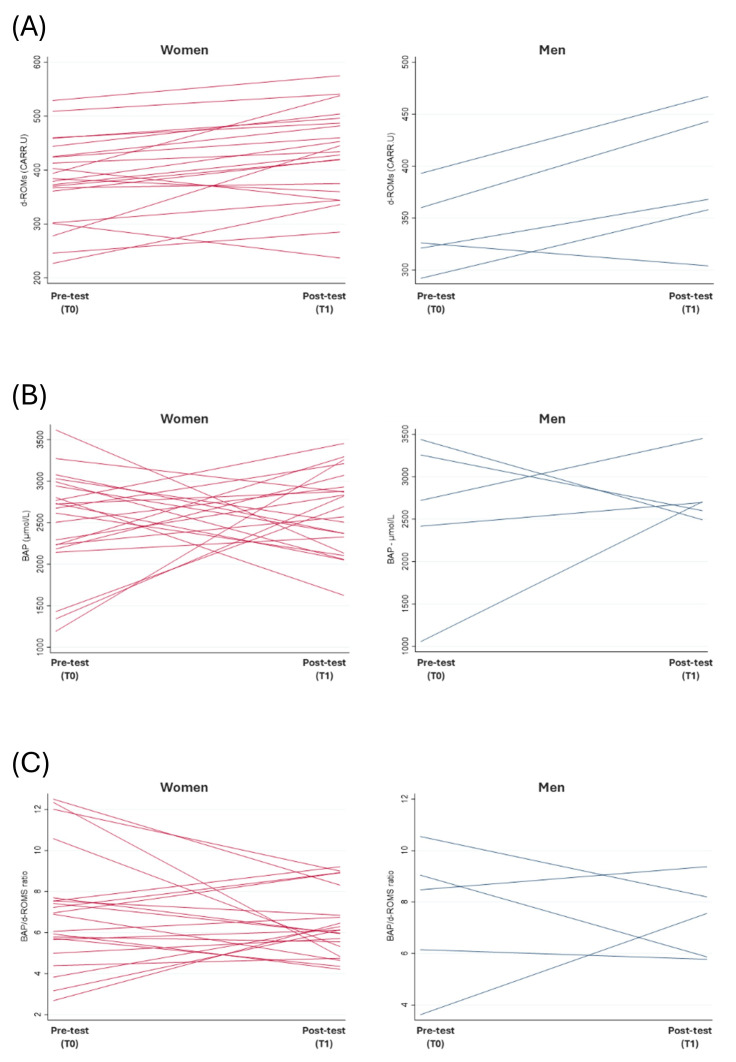
Changes in d-ROM values (**A**), BAP values (**B**), and BAP/d-ROM ratio (**C**) from pre-test assessment (T0) to post-test (T1) at individual level (each line represents an employee).

**Table 1 jfmk-10-00348-t001:** Sample’s physical activity levels and use of medicine and dietary supplements.

	Women (21)	Men (5)	Total (26)	Sign.
Regular Physical Activity	19 (90.5%)	5 (100.0%	24 (92.3%)	n.s. ^(a)^
No. of times/week—mean (range)	2.9 (1–7)	3.2 (2.4)	3.0 (1–7)	n.s. ^(b)^
Total min/week—mean (range)	195.8 (60–420)	228.0 (180–360)	202.5 (60–420)	n.s. ^(b)^
Use of medicine	9 (42.9%)	5 (100.0%)	14 (53.8%)	*p* < 0.05 ^(a)^
No. of medications—mean (range)	0.8 (0–5)	1 (1–1)	0.9 (0–5)	n.s. ^(b)^
Use of supplements	14 (66.7%)	2 (40.0%)	16 (61.5%)	n.s. ^(a)^

^(a)^ Fisher’s test; ^(b)^ Mann–Whitney rank-sum test.

**Table 2 jfmk-10-00348-t002:** Mass and metabolic risk measures over time in employees stratified by sex.

	Women (21)			Men (5)		
	T0	T1	Sign.	T0	T1	Sign.
Body Mass Index (kg/m^2^)	24.9 ± 4.4	24.8 ± 4.2	n.s. ^(a)^	25.3 ± 3.6	25.2 ± 3.3	n.s. ^(a)^
Chol TOT (≥200 mg/dL)	12 (57.1%)	6 (28.6%)	*p* ≤ 0.05 ^(b)^	1 (20.0%)	0 (0.0%)	n.s. ^(b)^
Chol HDL (<35 or >75 mg/dL)	1 (4.8%)	1 (4.8%)	n.s. ^(b)^	0 (0.0%)	0 (0.0%)	n.s. ^(b)^
Chol LDL (≥130 mg/dL)	12 (57.1%)	5 (23.8%)	*p* < 0.01 ^(b)^	1 (20.0%)	0 (0.0%)	n.s. ^(b)^
Triglycerides (≥130 mg/dL)	2 (9.5%)	1 (4.8%)	n.s. ^(b)^	1 (20.0%)	2 (40.0%)	n.s. ^(b)^
Fasting Glucose (≥110 mg/dL)	2 (9.5%)	0 (0.0%)	n.s. ^(b)^	0 (0.0%)	0 (0.0%)	n.s. ^(b)^

^(a)^ Wilcoxon matched-pairs signed-rank test; ^(b)^ McNemar’s test.

**Table 3 jfmk-10-00348-t003:** Physical fitness levels assessed by motor tests over time, stratified by sex.

	Women (15)			Men (5)		
	T0	T1	Sign.	T0	T1	Sign.
**One Leg Stand**						
High	8 (53.3%)	8 (53.3%)	n.s. ^(a)^	3 (60.0%)	5 (100.0%)	n.s. ^(a)^
Low/Mid	7 (46.7%)	7 (46.7%)		2 (40.0%)	0 (0.0%)	
**Shoulder/Neck Mobility**						
None or mild restriction (4–5)	6 (40.0%)	8 (53.3%)	n.s. ^(a)^	2 (40.0%)	2 (40.0%)	n.s. ^(a)^
Moderate or high restriction (1–3)	9 (60.0%)	7 (46.7%)		3 (60.0%)	3 (60.0%)	
**Handgrip**						
kg (mean ± standard deviation)	28.7 ± 6.1	30.0 ± 6.6	n.s. ^(b)^	48.0 ± 5.9	50.6 ± 11.0	n.s. ^(b)^
**Jump and Reach**						
High score (3–4° quartile)	9 (60.0%)	8 (53.3%)	n.s. ^(a)^	2 (40.0%)	4 (80.0%)	n.s. ^(a)^
Low score (1–2° quartile)	6 (40.0%)	7 (46.7%)		3 (60.0%)	1 (20.0%)	
**Dynamic Sit-up**						
High	10 (66.7%)	12 (80.0%)	n.s. ^(a)^	5 (100.0%)	5 (100.0%)	N/A ^(a)^
Low/Mid	5 (33.3%)	3 (20.0%)		0 (0.0%)	(0.0%)	
**2-Minute Step Test**						
% at risk (<65 steps)	1 (6.7%)	2 (13.3%)	n.s. ^(a)^	0 (0.0%)	0 (0.0%)	N/A ^(a)^
No. steps (mean ± standard dev.)	82.7 ± 16.8	85.5 ± 19.0	n.s. ^(b)^	87.6 ± 16.2	100.0 ± 21.8	n.s. ^(b)^ *p* = 0.0796

^(a)^ McNemar test for paired samples; ^(b)^ Wilcoxon matched-pairs signed-rank test data; N/A = not applicable.

**Table 4 jfmk-10-00348-t004:** Mean values of pro-oxidant (d-ROMs) and antioxidant status (BAP-test) measures and derived ratio stratified by time of observation and gender.

	Women (21)			Men (5)		
	T0	T1	Sign. ^(a)^	T0	T1	Sign. ^(a)^
d-ROMs (U.CARR)	383.2 ± 79.3	426.8 ± 87.3	*p* < 0.01 ^(a)^	338.4 ± 38.9	388.0 ± 66.4	n.s. ^(a)^ *p* = 0.0796
BAP (µmol/L)	2513.8 ± 624.6	2640.4 ± 488.2	n.s. ^(a)^	2578.2 ± 943.6	2789.2 ± 379.6	n.s. ^(a)^
BAP/d-ROM ratio	7.00 ± 2.8	6.40 ± 1.6	n.s. ^(a)^	7.6 ± 2.7	7.4 ± 1.5	n.s. ^(a)^

^(a)^ Wilcoxon matched-pairs signed-rank test.

**Table 5 jfmk-10-00348-t005:** Distribution of employees by level of pro-oxidant (d-ROM test) and antioxidant (BAP test) status and by oxidative risk categories [44] stratified by observation time and gender.

	Women (21)			Men (5)		
	T0	T1	Sign.	T0	T1	Sign.
**d-ROMs**						
Not high (<300 U.CARR)	3 (14.3%)	2 (9.5%)	n.s. ^(a)^	1 (20.0%)	0 (0.0%)	n.s. ^(a)^
High (≥300 U.CARR)	18 (85.7%)	19 (90.5%)		4 (80.0%)	5 (100.0%)	
**BAP**						
Optimal (≥2200 µmol/L)	16 (76.2%)	16 (76.2%)	n.s. ^(a)^	4 (80.0%)	5 (100.0%)	n.s. ^(a)^
Not optimal (<2200 µmol/L)	5 (23.8%)	5 (23.8%)		1 (20.0%)	0 (0.0%)	
**Oxidative risk**						
Optimal ox balance [d-ROMs ~ BAP ~]	1 (4.8%)	1 (4.8%)	n.s. ^(b)^	0 (0.0%)	0 (0.0%)	n.s. ^(b)^
Relative hyporesponsiveness [d-ROMs ↓ BAP ~]	2 (9.5%)	0 (0.0%)		0 (0.0%)	0 (0.0%)	
Absolute hypo-reactivity [d-ROMs ↓ BAP ↓]	0 (0.0%)	1 (4.8%)		0 (0.0%)	0 (0.0%)	
Relative stress ox [d-ROMs ~ BAP ↓]	0 (0.0%)	0 (0.0%)		1 (20.0%)	0 (0.0%)	
Stress ox potential [d-ROMs ↑ BAP ~]	13 (61.9%)	15 (71.4%)		4 (80.0%)	5 (100.0%)	
Absolute stress ox [d-ROMs ↑ BAP↓]	5 (23.8%)	4 (19.1%)		0 (0.0%)	0 (0.0%)	

^(a)^ McNemar test; ^(b)^ symmetry test; ~ similar; ↑ high levels; ↓ low levels.

**Table 6 jfmk-10-00348-t006:** Oxidative stress indices (BAP/d-ROM ratio and proportion of absolute oxidative stress [44]), metabolic risk, and health-related habits. Univariate analysis of whole sample (26 employees) at T0.

Metabolic Risk and Lifestyle Factors Whole Sample at T0 (26)	BAP/d-ROM Ratio Mean ± St Dev.		Sign.	Absolute Oxidative Stress %	Sign.
**Weight**	Normal	8.1 ± 2.7	*p* < 0.05 ^(a)^	23.1%	n.s. ^(b)^
	Overweight or Obesity	6.1 ± 2.4		15.4%	
**Total Cholesterol**	<200 mg/dL	8.7 ± 2.7	*p* < 0.01 ^(a)^	7.7%	n.s. ^(b)^
	≥200 mg/dL	5.5 ± 1.5		30.8%	
**LDL Cholesterol**	<130 mg/dL	8.7 ± 2.7	*p* < 0.01 ^(a)^	7.7%	n.s. ^(b)^
	≥130 mg/dL	5.5 ± 1.5		30.8%	
**Triglycerides**	<60 mg/dL	8.5 ± 3.1	*p* = 0.0719 ^(c)^	16.7%	n.s. ^(b)^
	60–130 mg/dL	7.0 ± 2.6		17.6%	
	≥130 mg/dL	4.8 ± 1.1		33.3%	
**Fasting Blood ** **Glucose**	<110 mg/dL	7.2 ± 2.8	n.s. ^(a)^	20.8%	n.s. ^(b)^
	≥110 mg/dL	6.0 ± 1.3		0.0%	
**Smoking**	Yes	5.0 ± 3.3	n.s. ^(c)^	50.0%	n.s. ^(b)^
	Ex	6.4 ± 2.3		12.5%	
	No	7.7 ± 2.8		18.8%	
**Alcohol** **(occasional)**	Yes	6.6 ± 2.0	n.s. ^(a)^	14.3%	n.s. ^(b)^
	No	7.7 ± 3.3		25.0%	
**Regular Physical ** **Activity**	Yes	7.3 ± 2.7	n.s. ^(a)^	17.7%	n.s. ^(b)^
	No	4.4 ± 0.8		50.0%	
**Medicine Use**	Yes	6.3 ± 2.6	*p* < 0.05 ^(a)^	14.3%	n.s. ^(b)^
	No	8.1 ± 2.6		25.0%	
**Supplement Use**	Yes	7.6 ± 2.8	n.s. ^(a)^	14.3%	n.s. ^(b)^
	No	6.5 ± 2.6		25.0%	

^(a)^ Mann–Whitney rank-sum test; ^(b)^ Fisher’s exact test; ^(c)^ Cuzick’s non-parametric trend test.

**Table 7 jfmk-10-00348-t007:** Oxidative stress indices (BAP/d-ROM ratio and proportion of absolute oxidative stress [44]) and physical fitness levels. Univariate analysis on whole sample (26 employees) at T0.

Performances Levels at Motor Tests Whole Sample at T0 (26)	BAP/d-ROM Ratio Mean ± St Dev.	Sign.	Absolute Oxidative Stress %	Sign.
**One leg stand**				
High	7.3 ± 2.4	n.s. ^(a)^	16.7%	n.s. ^(b)^
Low/Mid	7.0 ± 3.0		21.4%	
**Shoulder/neck mobility**				
No or light limitation (score 4–5)	6.8 ± 3.0	n.s. ^(a)^	25.0%	n.s. ^(b)^
Moderate or severe limitation (score 1–3)	7.4 ± 2.5		14.3%	
**Handgrip**				
Spearman’s Rho	0.242	n.s ^(d)^	-	
1° tertile	6.8 ± 3.1	n.s. ^(c)^	36.3%	n.s. ^(b)^
2° tertile	6.1 ± 2.0		16.7%	
3° tertile	8.2 ± 2.6		0.0%	
**Jump and reach**				
3–4° quartile	7.04 ± 2.7	n.s. ^(a)^	14.3%	n.s. ^(b)^
1–2° quartile	7.2 ± 2.9		25.0%	
**Dynamic sit-up**				
High	7.7 ± 2.5	*p* = 0.0668 ^(a)^	11.1%	n.s. ^(b)^
Low/Mid	5.8 ± 3.0		37.5%	
**2-Minute Step Test**				
Risk (<65 elevations)	3.9 ± 1.3	*p* < 0.01 ^(a)^	100.0%	*p* < 0.001 ^(b)^
No risk (≥65 elevations)	7.7 ± 2.5		4.5%	
Spearman’s Rho	0.404	*p* < 0.05 ^(d)^	-	
1° tertile	6.1 ± 2.7	*p* = 0.0519 ^(c)^	30.8%	n.s. ^(b)^
2° tertile	8.1 ± 2.9		0.0%	
3° tertile	8.1 ± 2.2		16.7%	

^(a)^ Kruskall–Wallis test; ^(b)^ Fisher’s exact test; ^(c)^ Cuzick’s non-parametric trend test; ^(d)^ significance test on rho in Spearman’s correlation.

## Data Availability

The raw data supporting the conclusions of this article will be made available by the authors upon request.

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
