# Peer review of "Impact of a 24-Week Workplace Physical Activity Program on Oxidative Stress Markers, Metabolic Health, and Physical Fitness: A Pilot Study in a Real-World Academic Setting"

_jfmk, 2025, doi:10.3390/jfmk10030348_

Round 1
Reviewer 1 Report
Comments and Suggestions for Authors
This manuscript presents a well-structured and thorough investigation, with a clear and coherent exposition across the introduction, methods, results, and discussion sections. The results are explained in detail and supported by a solid analysis of the implications and physiological responses to the intervention, particularly emphasising the role of physical exercise in improving oxidative status. While the study is well-conducted and scientifically sound, several minor revisions could enhance the logical flow between concepts and allow certain aspects to be explored in greater depth. The topic is highly relevant, and the work makes a valuable contribution to understanding the effects of physical activity in the workplace. It has the potential to serve as a helpful reference for future interventions in occupational settings and for promoting healthier lifestyles.
Line 53–54: It may be helpful to mention the link between sedentary working conditions and the incidence of chronic diseases such as diabetes or cardiovascular disease, underlining the importance of workplace interventions.
Line 70–85: The text is clear and well-written, but could benefit from greater synthesis to avoid repetition and improve clarity.
Line 97–99: Consider adding a brief note on how this research contributes specifically to the field and what differentiates it from previous studies. For example, indicate whether the program stands out due to a particular methodological approach or an innovative design.
Line 229–230: Please verify the accuracy of the data presented in the table.
Line 364–366: It would be helpful to clarify how your findings relate to the conclusions of previous studies. For example: These findings are consistent with previous research suggesting that regular exercise can improve the lipid profile, but they differ from approaches that...
Line 429–430: You could elaborate on how the absence of a control group may have limited the ability to attribute changes solely to the intervention, and suggest alternatives for future research (e.g., inclusion of a randomised control group).
Line 433–435: Consider specifying how these factors influenced statistical power. For example, gender distribution or age heterogeneity may have contributed to high variability, thereby reducing the reliability of the conclusions.
Line 439–441: You could add a brief explanation of the type of population that would be most appropriate for future studies, such as samples with more homogeneous demographic characteristics or a more detailed analysis of participants’ health backgrounds.
Reviewer 2 Report
Comments and Suggestions for Authors
Please, find my comments in the attached PDF.

Reviewer 3 Report
Comments and Suggestions for Authors
A 24-Week Workplace Physical Activity Program in an Academic Community: Impact on Oxidative Redox Status and Metabolic and Physical Fitness as Health Outcomes in a Real-World Setting
This is, it truly is, a thoughtfully planned and pertinent study! It dives deep into a significant area that deserves attention: how a workplace workout scheme affects things like oxidative stress, our metabolism health, and how physically fit we are. The way the research was set up makes good sense for a real-life scenario. Overall, the methods used seem solid enough too. And the results showing better lipid profiles, and how they may interact with statin treatment, those are worth a look. Yet, a few crucial points need considerable work. We need clearer explanations, a more robust statistical picture, and more honestly consider all those tricky limitations so we can get this up to publishing level.
Major comments
Abstract:
Objective: The objective should be more precise. Instead of "evaluate the impact," specify the direction of the expected change (e.g., "to evaluate the improvements in..."). Also, explicitly mention the analysis of the interaction with statin treatment, as it is a major finding.
Results: The statement "d-ROMs values increased significantly" is potentially misleading without immediate context. It must be explicitly framed as a likely acute adaptive response to exercise, not a negative outcome. The results for the statin subgroup are a key finding and deserve a mention in the abstract.
Conclusion: The conclusion should be more direct and confident, summarizing the main take-homes: The program improved metabolic health and maintained redox balance despite increased ROS production. Furthermore, statin use may attenuate the antioxidant response to exercise, a novel finding requiring further study.
Introduction:
Gap Identification: Though well defined, the gap deserves a tad more punch. Specifically highlighting, despite the well-documented benefits of WPAP, how it really impacts the dance of oxidative stress using the BAP/d-ROMs ratio, within a true academic scenario, is barely touched on, particularly how pre-existing drugs like statins tweak things needs more exploration.
Reference Update: To spice up the background, chuck in some fresh reviews about oxidative stress linked to exercise.
Methods:
Sample Size Justification: Power calculations, or a direct explanation, conceding the study's pilot phase, crucial for the realistic approach is mandatory. This preempts criticism regarding any lacklustre findings.
Statin Subgroup Analysis: Reasons for splitting statin takers are needed in the methods, no jumping ahead to results. Clarify that, with statins impact on mitochondrial function and CoQ10 known, also, reference Hargreaves et al. , 2020, for example, and planned a subgroup analysis.
Intervention Adherence: Mention adherence figures attendance rates per participant, essential when looking at the result.
Results:
Statistical Narrative: Refrain from reading too much into unremarkable shiftslike "small gains" or a "tiny dip". Report data factually: "A not-important rise seen".
Tables: Table 3 got wacko sample numbers (n=15 females, n=5 males) than the full group (n=21 females, n=5 males). Dig in, uncover the reason for these differences.
Figure 1: Figure captions 1A and 1B needs better details, right in the visual or nearby text. Define the groups so its easy to understand. Error bars needs clarification like, "±SEM" or something like "±SD".
Discussion:
Understanding rising d-ROMs: This is pivotal. You really must champion the increase in d-ROMs as a potentially beneficial, hormetic adaptation to the workout program, not an issue with the study. Reference studies on oxidative eustress provoked by exercise such as Radak et al 2008, would be good. The steady BAP/d-ROMs ratio provides you're the sturdiest supporting evidence.
Statin Findings: This result sparks interest! Investigate this much further. Delve into the suggested mechanics, say, statins possibly lowering CoQ10, a vital part of the mitochondrial electron chain and also antioxidant. Add appropriate literature, example, Zinellu & Mangoni, 2021.
Limitations: That part, it is okay, however it has to be a tad bolder. State outright, the lack of a control group does mean we cannot say everything that changed was down to the intervention only think about seasonal things or reversion to the middle value. The small, mixed sample size complicates generalizing plus statistically managing variables, also.
Minor comments
Title: Consider simplifying slightly: "...Impact on Oxidative Stress Markers, Metabolic Health, and Physical Fitness..."
Keywords: Add "statin" as a keyword given the findings.
Section 3.1: "auto-referred" should be changed to "self-reported."
Throughout: Perform a thorough proofread for minor grammatical errors and typos (e.g., "upper limp" should be "upper limb" in section 2.8).
My honest thoughts are, this paper's got real important findings stemming from a hands-on intervention study. Those good changes in lipid profiles, and the super interesting link between exercise-related stress and statin treatment, really add to what we know. Still, the manuscript needs a whole lotta work - especially how they explain the oxidative stress stuff, plus a much clearer explanation of how they did the study, and also, a beefier discussion really tackling the downsides.
I recommend major revision. If they fix what I mentioned, this manuscript has a decent shot at getting published in the Journal of Functional Morphology and Kinesiology.
Reviewer 4 Report
Comments and Suggestions for Authors
Please indicate whether or not the participants signed the informed consent form.
Lifestyle and dietary changes could have influenced lipid profile modifications.
Generally speaking, when people undergo a controlled or systematic exercise program, they tend to modify their lifestyle, including their diet. How did the authors monitor these changes?
Due to the lack of a control group, it is not possible to determine whether the treatment had an effect on its target variables.
Therefore, this study is only a pilot study: Impact of a 24-Week Workplace Physical Activity Program on Oxidative Redox Status, Metabolic and Physical Fitness: A Pilot Study.
Round 2
Reviewer 2 Report
Comments and Suggestions for Authors
I want to congratulate the authors on their work to improve the manuscript. I now believe that the manuscript was significantly improved and ready for publication.
Reviewer 3 Report
Comments and Suggestions for Authors
No comments to add.
Reviewer 4 Report
Comments and Suggestions for Authors
The authors did a good job and responded to suggestions and inconsistencies. I have no further questions.